# Mortality and Survival Factors in Patients with Moderate and Severe Pneumonia Due to COVID-19

**DOI:** 10.3390/healthcare11070932

**Published:** 2023-03-23

**Authors:** Evelyn Galindo-Oseguera, Rodolfo Pinto-Almazán, Alfredo Arellano-Ramírez, Gilberto Adrián Gasca-López, María Esther Ocharan-Hernández, Claudia C. Calzada-Mendoza, Juan Castillo-Cruz, Erick Martínez-Herrera

**Affiliations:** 1Escuela Superior de Medicina, Instituto Politécnico Nacional, Plan de San Luis y Díaz Mirón, Ciudad de México 11340, Mexico; dr.evelyngalindooseguera@gmail.com; 2Sección de Estudios de Posgrado e Investigación, Escuela Superior de Medicina, Instituto Politécnico Nacional, Plan de San Luis y Díaz Mirón, Ciudad de México 11340, Mexico; rodolfopintoalmazan@gmail.com (R.P.-A.); estherocharan@hotmail.com (M.E.O.-H.); cccalzadam@yahoo.com.mx (C.C.C.-M.); 3Hospital Regional de Alta Especialidad de Ixtapaluca, Ixtapaluca 56530, Mexico; alfredo_cancer82@hraei.gob.mx (A.A.-R.); gadriangasca1@yahoo.com.mx (G.A.G.-L.); 4Efficiency, Quality, and Costs in Health Services Research Group (EFISALUD), Galicia Sur Health Research Institute (IIS Galicia Sur), SERGAS-UVIGO, 36213 Vigo, Spain

**Keywords:** SARS-CoV-2, COVID-19, risk factors, survival factors, mortality factors

## Abstract

During the pandemic, some mortality-related factors were age, sex, comorbidities (obesity, diabetes mellitus, and hypertension), recovery time, hospitalizations, and biochemical markers. The present work aimed to identify the mortality and survival factors in adults with moderate and severe pneumonia due to COVID-19 during the first and second waves of the pandemic in Mexico at a third-level hospital (High-Specialty Regional Hospital of Ixtapaluca (HRAEI), Ixtapaluca, Estado de Mexico, Mexico). A database was generated using information from the electronic clinical records of patients hospitalized from December 2021 to August 2022. Survival analysis was performed associating age, sex, longer recovery times, and some drugs. The risk factors found were age in the patients between 40 and 60 years (OR = 1.70), male sex (OR = 1.53), the presence of comorbidities (OR = 1.66) and hypertension (OR = 2.19), work occupation (construction workers OR = 5.22, factory workers OR = 3.13, unemployed OR = 2.93), the prehospital use of metamizole sodium (OR = 2.17), cough (OR = 1.73), and in-hospital oxygen therapy (reservoir mask OR = 6.6). The survival factors found in this study were working in the healthcare field (OR = 0.26), the prehospital use of certain medications (paracetamol OR = 0.65, dexamethasone OR = 0.55, and azithromycin OR = 0.47), presenting ageusia (OR = 0.5) and hyporexia (OR = 0.34), and the time using in-hospital oxygen therapy (device 1 OR = 0.72). Prehospital treatment needs to be reevaluated as dexamethasone and azithromycin proved to be protective factors. Likewise, providing aggressive oxygen therapy during hospital admission decreased mortality risk.

## 1. Introduction

In December 2019, a new coronavirus emerged called severe acute respiratory syndrome coronavirus 2 or SARS-CoV-2 that has left a total of 651,918,408 cumulative cases worldwide and a total of 6,656,601 deaths up to December 2022. It is worth mentioning that some studies have reported that the mortality recorded during the first waves of COVID-19 could be underestimated as several patients died before having a positive real time polymerase chain reaction (RT-PCR) test or false-negative results [1]. In the Americas, 2,885,816 deaths and 185,198,746 confirmed cases have been reported thus far, with Mexico occupying the fifth place with 7,165,257 cases and third in deaths with 330,699 fatalities [2,3].

During this COVID-19 pandemic, certain features associated with COVID-19 mortality have been identified, such as age over 60 years, male sex, comorbidities such as obesity, overweight, diabetes mellitus (DM), high blood pressure or hypertension (HBP), among others, as well as other biochemical parameters such as high levels of C-reactive protein (CRP), lactic dehydrogenase (DHL), and ferritin. In addition, neutrophilia, lymphocytopenia, and thrombocytopenia can also occur. Other identified risk factors are recovery time, hospital stays, and the presence of some symptoms (such as fever, dyspnea, and cough). On the other hand, the survival factors identified are age less than 60 years, headaches, and the absence of comorbidities [4,5,6,7,8,9,10].

Likewise, in Mexico, previous reports show that the population aged 20 years and over has a high prevalence of diabetes mellitus (15.6%), hypertension (30.2%), cardiovascular disease (1.7%), chronic lung disease (2.1%), dyslipidemia (19.5%), and overweight and obesity (75.2%). In addition, according to the ENSANUT 2020, it was reported that 16.8% of people consume tobacco and 54.3% alcohol [11,12,13,14,15,16,17]. Finally, the lack of physical activity increased by 29.1% due to the lockdown, favoring the development or prevalence of chronic diseases [11,13,16].

Other factors to consider regarding mortality and patient survival during the pandemic are socioeconomic and demographic factors. With the confinement, a slowdown in some productive activities was observed, increasing poverty and unemployment. This caused poor access to anti-contagion measures, which increased exposure to the virus in the most vulnerable sectors [18].

The present work aimed to identify the mortality and survival factors in adults with moderate and severe pneumonia due to COVID-19 during the first and second waves of the pandemic through the analysis of sociodemographic characteristics, symptoms, body mass index (BMI), comorbidities, prehospital treatment, and biochemical determinations upon hospital admission.

## 2. Materials and Methods

This was an observational, analytical, retrospective, longitudinal, and cohort study. The data collected were age, sex, BMI, education, occupation (merchants, housewives, unspecified occupation, general employees, carriers, healthcare personnel, manufacturing workers, administrative personnel, construction workers, unemployed, police officers or private security, farmers, lawyers, teachers, funeral service workers, students, retired people, and lastly engineers), known comorbidities at the time of hospital admission and their recovery time, the evolution time from the onset of symptoms to the hospital admission due to SARS-CoV-2, the medications used before hospitalization, the length of hospital stay, and the use of in-hospital oxygen therapy (device 1 = the first device used from the hospital admission to the adjustment of oxygen supply, including nasal tips, reservoir mask, high flow tips, and mechanical ventilation; device 2 = the next device used after adjusting the oxygen therapy). A database was consolidated using information from the electronic clinical records of patients hospitalized at the High-Specialty Regional Hospital of Ixtapaluca (HRAEI), Ixtapaluca, Estado de Mexico, Mexico, from December 2021 to August 2022. The third-level hospital has 262 hospital active beds and 30 inactive beds, which increased to 330 during the pandemic. It is essential to mention that the hospital was exclusively intended to care for COVID-19 patients during the first three waves of the pandemic. This research project was endorsed by the Research and Research Ethics Committees of the HRAEI (NR-32-2021 and NR-CEI-HRAEI-32-2021). The sample size was calculated using the G*power program with a power of 80%, an alpha error of 0.05%, and an odds ratio of 1.5%, which yielded 407 subjects needed for this study. The inclusion criteria were patients aged 20 to 60 who presented the disease from February 2020 to April 2021 with a positive PCR result for SARS-CoV-2, required hospitalization, and had laboratory tests conducted at admission. The non-inclusion criteria were patients with mild pneumonia previously treated at other hospitals. The elimination criteria were patients with voluntary discharge and records with missing data.

### Statistical Analysis

Statistical analysis was used to compare the data collected from patients who died and living patients. Using the R studio^®^ 4.1.1 software, normality tests such as Kolmogórov–Smirnov, asymmetry, kurtosis, and q-q plots were performed, allowing us to choose between the *t-Student* test or X^2^ square, as well as risk estimation based on the odds ratio obtained with bivariate and multivariate logistic regression models. The survival curve and risk estimation were calculated with Kaplan–Meier models.

## 3. Results

The studied population consisted of 417 subjects, of which 33% were women and 66% were men, with a mean age of 47 and an age range between 20 and 60. The mean height and weight were 1.64 m and 84.7 kg, respectively. Regarding the BMI, grade 1 obesity was observed in 32%, overweight in 25.7%, grade 2 obesity in 12%, and the average weight in 9% of the population. Regarding the level of education, the following order was observed: 18.3% completed elementary school, 34.6% completed middle school, 26.9% completed high school, and 14.4% had a bachelor’s degree. Likewise, the jobs and occupations were identified as merchants (22.8%), housewives (7.6%), unspecified occupations and general employees (15.9% each), carriers (6.3%), healthcare personnel (4%), factory workers (3%), administrative personnel (2.8%), construction workers (2.6%), unemployed (2.4%), police officers or private security (1.9%), farmers (1.4%), lawyers (1.2%), teachers (0.7%), funeral service workers students and retired people (0.4% each), and lastly engineers (0.2%). The mean number of days was 10 for the recovery period, 7.5 for the hospital stay, and 17.5 for the total time of the disease. In addition, 54.6% had at least one comorbidity.

When studying whether there was an association between sex, occupation, comorbidities, symptoms, prehospital treatment, in-hospital oxygen therapy, and mortality in patients with moderate or severe pneumonia due to COVID-19, the following results were observed. Regarding sex, an OR = 0.47 (95%; CI [0.30–0.72]; *p* < 0.05) was obtained, with women having half the risk compared to men. As for the occupation, ordered by the greatest risk, the results are as follows: construction workers with an OR = 5.22 (95%; CI [1.37–2.55]; *p* < 0.05), unemployed subjects with an OR = 2.93 (95%; CI [0.76–1.24]; *p* = 0.06), and carriers with an OR = 2.28 (95%; CI [0.91–5.79]; *p* = 0.07). However, some occupations showed a lower risk, such as housewives with an OR = 0.51 (95%; CI [0.30–0.82]; *p* < 0.05) and healthcare workers with an OR = 0.26 (95%; CI [0.03–1.02]; *p* = 0.09). On the other hand, having at least one comorbidity resulted in an OR = 1.66 (95%; CI [1.12–2.47]; *p* < 0.05). The highest risk was the intake of illegal drugs (OR = 4.8; 95%; CI [1.15–32.68]; *p* = 0.07), HBP (OR = 2.19; 95%; CI [1.39–3.49]; *p* < 0.05), alcoholism (OR = 1.58; 95%; CI [1.00–2.51]; *p* < 0.05), and DM (OR = 1.42; 95%; CI [0.90–2.24]; *p* = 0.15).

Concerning the symptoms due to COVID-19, those with the highest risk were dyspnea (OR = 0.93; 95%; CI [0.87–1.00]; *p* = 0.06), myalgias (OR = 0.92; 95%; CI [0.86–0.98; *p* < 0.05), arthralgias (OR = 0.91; 95%; CI [0.85–0.97]; *p* < 0.05), headaches (OR = 0.9; 95%; CI [0.83–0.96]; *p* < 0.05), and ageusia (OR = 0.87; 95%; CI [0.81–0.93]; *p* < 0.05).

As for the prehospital treatment of COVID-19, the one with the highest risk was metamizole sodium (OR = 2.57; 95%; CI [1.40–4.86]; *p* < 0.05). Those with protective effects included paracetamol (OR = 0.65; 95%; CI [0.43–0.97]; *p* < 0.05), azithromycin (OR = 0.47; 95%; CI [0.30–0.74; *p* < 0.05), ceftriaxone (OR = 0.47; 95%; CI [0.42–0.92]; *p* < 0.05), enoxaparin (OR = 0.47; 95%; CI [0.19–1.05]; *p* = 0.07), and steroids (OR = 0.55; 95%; CI [0.34–0.89]; *p* < 0.05), specifically dexamethasone (OR = 0.49; 95%; CI [0.30–0.76]; *p* < 0.05).

Regarding in-hospital oxygen therapy, the highest risk was for nasal tips (OR = 0.15; 95%; CI [0.07–0.29]; *p* < 0.05) and the reservoir mask (OR = 6.6; 95%; CI [3.32–1.46]; *p* < 0.05). See Table 1.

The following analysis was performed when studying whether there was an association between mortality in subjects with moderate and severe pneumonia due to COVID-19 and age, the progression of the disease over time, total hospital stays, stays per service, symptomatology progression over time, dose, the length of use of prehospital treatment, and the amount of time using intrahospital oxygen therapy. The mean age in years was 45.4 for living patients and 49.1 for deceased patients (OR = 1.04; 95%; CI [1.02–1.07]; *p* < 0.05). The age group with the highest risk was between 40 and 60 years old (OR = 1.70; 95%; CI [1.03–2.01]; *p* < 0.05).

The mean length of hospitalizations was 9.56 days for living patients and 4.6 for deceased patients (OR = 0.75; 95%; CI [0.69–0.79]; *p* < 0.05), and urgent care had a mean of 0.7 days for living patients and 0.8 for deceased patients (OR = 2.32; 95%; CI [1.27–4.43]; *p* < 0.05). As for the recovery time, the mean was 10.5 days for living patients and 8.5 for deceased patients (OR = 0.89; 95%; CI [0.84–0.94]; *p* < 0.05).

With reference to the duration of symptoms, the mean in days for headaches was 9.1 for living patients and 7.5 for deceased patients (OR = 0.9; 95%; CI [0.83–0.96; *p* < 0.05), 9.5 for arthralgias for living patients and 8.1 for deceased patients (OR = 0.91; 95%; CI [0.85–0.97]; *p* < 0.05), and 9.4 for myalgias for living patients and 8.2 for deceased patients (OR = 0.92; 95%; CI [0.86–0.98]; *p* < 0.05). For asthenia or adynamia, a mean of 9.4 was obtained for living patients and 7.6 for deceased patients (OR = 0.87; 95%; CI [0.81–0.93]; *p* < 0.05). For dyspnea, a mean of 3.5 was obtained for living patients and 2.9 for deceased patients (OR = 0.93; 95%; CI [0.87–1.00]; *p* < 0.05).

Concerning prehospital treatment, only oseltamivir was associated with mortality. A total of 49 subjects received a dose prior to hospital admission. The mean was 129 mg for living patients and 220 mg for deceased patients (OR = 1.02; 95%; CI [1.01–1.06]; *p* < 0.05). The time of use in days of some drugs proved to be a protective factor (paracetamol $\bar{x}$ living = 6; $\bar{x}$ deceased = 4.8 (OR = 0.85%; 95%; CI [0.72–0.99]; *p* ˂ 0.05) azithromycin $\bar{x}$ living = 2.9; $\bar{x}$ = deceased = 4 (OR = 0.74; 95; CI [0.55–0.96]; *p* ˂ 0.05)).

As for the time of in-hospital oxygen therapy, the mean time in days for device 1 was 4.9 for living patients and 2.6 for deceased patients (OR = 0.72; 95%; CI [0.65–0.79]; *p* < 0.05). On the other hand, for device 2, the mean was 3.42 for living patients and 2.17 for deceased patients (OR = 0.91; 95%; CI [0.84–0.98]; *p* < 0.05). See Table 2.

The overall survival curve of subjects with moderate or severe pneumonia due to COVID-19 who required hospitalization had a median of 23 days (95%; CI = 20–NA) with an overall survival rate of 42%, as presented in Figure 1A. The survival curve, according to sex, had a value of *p* < 0.05; survival for women was 55%, and it was not possible to calculate the median since more than 50% survived (95%; CI = 23–NA). On the other hand, survival for men was 36% and the median was 19 days (95%; CI = 18–25), as shown in Figure 1B. The survival curve of subjects who required hospitalization related to the presence or absence of comorbidities had a *p* < 0.05 value, with a survival rate of 47% for subjects with no comorbidities and a median of 32 days (95%; CI = 22–NA), while those with comorbidities had a survival rate of 40% with a median of 19 days (95%; CI = 17–NA), as shown in Figure 1C. The survival curve related to the presence or absence of HBP had a *p* < 0.05 value, with a 47% survival rate for subjects with no HBP and a median of 32 days (95%; CI = 23–NA), while those who presented HBP had a 30% survival rate with a median of 17 days (95%; CI = 14–22), as illustrated in Figure 1D. The survival curve related to the presence or absence of drug addiction, that is, the intake of illegal drugs, had a value *p* = 0.07, close to significance; survival for subjects with no drug addiction was 43% with a median of 25 days (95% CI = 21–NA), while those who consumed illegal drugs had a survival rate of 30% with a median of 18 days (95%; CI = 12–NA), as shown in Figure 1E. The survival curve related to the presence or absence of DM had a *p* < 0.05 value; survival for the absence of DM was 44% with a median of 26 days (95%; CI = 22–NA), while those with DM had a survival rate of 30% with a median of 18 days (95%; CI = 16–NA), as seen in Figure 1F. The survival curve related to the presence or absence of cough as a COVID-19 symptom had a *p* < 0.05 value; survival for patients with no cough was 49% with a median of 32 days (95%; CI = 25–NA), while those with a cough had a survival rate of 41% with a median of 20 days (95%; CI = 18–26), as shown in Figure 1G. The curve related to the presence or absence of hyporexia as a COVID-19 symptom had a value *p* = 0.05, with 41% survival for the absence of hyporexia and a median of 23 days (95%; CI = 19–NA), while those with hyporexia had a survival rate of 71%; the median, in this case, was not calculated since survival was greater than 50% (95%; CI = 26–NA), as seen in Figure 1H. The survival curve related to the presence or absence of dyspnea as a COVID-19 symptom had a value *p* = 0.5, with a 75% survival rate for the absence of this symptom and a non-quantifiable median as survival was greater than 50% (95%; CI = NA) Figure 1I Subjects with hyporexia had a survival rate of 42% with a median of 23 days (95%; CI = 20–NA), as shown in Figure 1. The survival curve related to the presence or absence of asthenia and adynamia as COVID-19 symptoms had a value *p* < 0.05, with a 54% survival rate for patients with no symptoms and a non-quantifiable median since survival was greater than 50% (95%; CI = 23–NA), while those with asthenia and adynamia had a survival rate of 35%, with a median of 22 days (95%; CI = 18–NA), as shown in Figure 1J. The survival curve associated with metamizole sodium as a prehospital treatment for COVID-19 had a *p* < 0.05 value; survival for subjects who did not use metamizole sodium was 44% with a median of 26 days (95%; CI = 22–NA), while those who used metamizole sodium had a survival rate of 26%, with a median of 17 days (95%; CI = 14–22), as presented in Figure 1K. The survival curve related to the use of steroids as prehospital treatment for COVID-19 had a *p* < 0.05 value; survival for subjects who did not use steroids was 37% with a median of 22 days (95%; CI = 19–NA), while those who used steroids had a 58% survival rate; it was not possible to calculate the median since survival was greater than 50% (95%; CI = 25–NA), as seen in Figure 1L.

## 4. Discussion

The HRAEI is a third-level hospital that pertains to the secretary of health; however, during the pandemic, by presidential decree, all patients diagnosed with COVID-19 could receive services at any public institution, even without benefits. In Mexico, many patients died during the first two waves due to difficulties being transferred to medical units, the saturation of health institutions, and the lack of supplies [18,19,20].

In the present study, the average age of patients mostly related to mortality was 40 to 60 years. Accordingly, O’Driscoll et al. reported in their meta-analysis of 45 countries that a logarithmic relationship exists between age and mortality, which is higher in people aged 30 to 65. Therefore, age is a non-modifiable factor associated with COVID-19 mortality, which increases after 40 years of age. The age group of 40 to 60 years had the highest prevalence in our country [21]. Likewise, Lutz et al. reported that aging is accompanied by a lower immune response to infections due to the immunosenescence process that affects both the adaptive and innate immune responses, preventing the body from responding adequately to SARS-CoV-2 infection [22]. Furthermore, the physiological changes at the respiratory level that occur with aging and that have been identified from the age of 35, such as a decrease and slowing of cilia, the flattening of alveoli, the decreased muscle mass of the respiratory muscles, and a loss of lung elasticity, together with the exposure to environmental pollution, gases, dust, tobacco smoke, bacteria, and viruses present in ambient air, have been associated with the development of severe COVID-19 forms in people over 30 years of age [23].

In this study, an increased risk was observed in the male sex. Consistent with the results of the present study, Ghadi et al., Gebhard et al., and Ueyama et al. reported in their meta-analyses that men had a less effective immune response than women, favoring the development of severe forms of the disease, and therefore a higher percentage of hospitalizations [24,25,26,27].

This study observed that some occupations, such as housewives and healthcare personnel, were protective factors. Most of the female population was engaged in domestic activities, and those with different occupations modified their enterprises to be able to work from home, thus avoiding exposure to virus carriers [28,29]. Likewise, being a healthcare worker was a protective factor. Although they had greater exposure to the disease, they had a better understanding of its forms of contagion and a better economic income, allowing them to acquire adequate personal protection measures. In contrast, occupations such as construction workers, factory workers, carriers, and the unemployed population were associated with increased mortality from COVID-19 since they had greater exposure to people infected with SARS-CoV-2. Moreover, because these are poorly remunerated jobs, it leads to an economic vulnerability that impacts both the purchase of protective equipment and access to health services [16,28,29]. According to the United Nations (UN), economic vulnerability during the pandemic, coupled with the difficulty of accessing basic services such as water and a higher level of overcrowding, led to the acquisition of the disease [30,31].

On the other hand, it was observed that about half of the patients in this study presented grade I and II obesity, which was positively associated with increased mortality. Similarly, Halem et al., who described the clinical characteristics, treatments, and complications of 319 patients with COVID-19 in a Hospital in Belgium, observed similar percentages (40% overweight and 20% obesity) to the present study [9].

As for comorbidities, we observed an association between the number of comorbidities and mortality from COVID-19. Among the comorbidities, the most related to increased mortality were HBP, alcoholism, and drug addiction. Following the results of the present study, several authors reported that HBP, cardiovascular diseases, diabetes mellitus, chronic obstructive pulmonary disease, and cancer are strongly related to mortality caused by this virus [5,9,32]. Salazar et al. mentioned that the expression of angiotensin-converting enzyme 2 (ACE2) receptors in the heart and blood vessels facilitates infection by SARS-CoV-2. The virus injures the myocardium via the overexpression of tumor necrosis factor (TNF), which could accelerate or perpetuate the elevation of blood pressure figures [33]. In addition, there is scientific evidence associating the lockdown with increased LDL concentration, presenting cardiovascular risk induced by the lack of physical activity [34]. Likewise, the Pan American Health Organization (PAHO) reported that using alcohol and drugs weakens the immune response, increasing the risk of infection and the severity of this disease. During the pandemic, these substances were consumed more [35,36,37]. Interestingly, unlike other research, in the present study, DM was not a risk factor since the highest prevalence of this disease is found in patients over 60 years of age [12,15,16,17].

Regarding the time of evolution, the total time of the disease, total hospital stay, and per service in this study, it was observed that they were shorter in patients who died, usually in less than ten days, with the highest mortality reported in the emergency department. Accordingly, Grasselli et al., in a cohort study of 4209 patients in the intensive care unit (ICU), reported that patients with survival had evolution times and hospital stays greater than ten days [38]. This may be because severe forms of COVID-19 present the cytokine storm, a dysregulated inflammatory process that leads to multiple organ failure (MOF), producing death rapidly [38,39,40,41,42].

Due to the poor knowledge of COVID-19 during the first waves of the pandemic, drug treatment was administered empirically and is still being studied [43,44,45,46]. Concerning prehospital treatment, it was observed in this work that the time of use and doses of paracetamol, dexamethasone, ceftriaxone, enoxaparin, and azithromycin were protective factors. In contrast, the use of metamizole sodium proved to be a risk factor. According to the Clinical Guidelines for the Treatment of COVID-19 in Mexico and the World Health Organization (WHO) guidance for clinical management of COVID-19, dexamethasone and enoxaparin were beneficial in moderate and severe forms requiring supplemental oxygen use. At the same time, non-steroidal anti-inflammatory drugs (NSAIDs) are indicated in mild forms, and antibiotic use is only prescribed for bacterial coinfection [47,48,49]. The RECOVERY study, a randomized clinical trial that investigated dexamethasone 6 mg every 24 h orally or intravenously for ten days, compared mortality at 28 days and reported therapeutic effects in patients who required invasive or non-invasive oxygen therapy, and no benefits were observed in subjects with a mild disease without supplemental oxygen; steroids regulated the inflammatory process and reduced the risk of respiratory failure [50]. As for antibiotics, some studies, such as those by Ishaqui et al. and Kakeya et al., support the use of azithromycin during the first hours of the diagnosis of viral pneumonia including COVID-19 [51,52]. Other clinical trials show improvements in critically ill patients using this drug, reducing invasive oxygen therapy time. The latter could be explained by the immunomodulatory action of cytokines released after treatment with azithromycin, cephalosporins, and fluoroquinolones [51,52,53,54,55,56,57,58,59]. During COVID-19, hypercoagulability, endothelial injury, and atherosclerosis have been described, predisposing to thrombosis and coagulopathies. Thromboprophylaxis (enoxaparin) helps reduce the risk of presenting these complications [38,39,40,41,42]. Regarding the use of NSAIDs, the mechanism that reduces the risk of mortality is unknown; however, it could be related to the safety profile they offer for cardiovascular, hemorrhagic, and renal risk, especially in older adults or people with multiple comorbidities [45]. In contrast, metamizole sodium has been associated with neutropenia and agranulocytosis by activating T lymphocytes that eliminate neutrophils in SARS-CoV-2 infection. Therefore, this drug could reduce the immune response, increasing mortality [60].

In this study performed during the first two waves of the pandemic, in-hospital oxygen therapy on admission was associated with COVID-19-related mortality. During this time, the population feared seeking medical help during the first stage of the disease, thus arriving at hospitals with severe hypoxemia. According to Ospina-Tascón et al., who conducted a randomized clinical trial in three hospitals in Colombia and evaluated the effect of conventional oxygen therapy, high-flow therapy, and mechanical ventilation, using high-flow therapy from hospital admission reduced the risk of complications and the need for mechanical ventilation compared to conventional oxygen multistep therapy [61]. In turn, Jiang and Wei reported that initial aggressive oxygen therapy in patients with COVID-19 reduces mortality and the likelihood of complications, and that high-flow cannulas are the best option prior to mechanical ventilation [62].

The High-Specialty Regional Hospital of Ixtapaluca, Estado de Mexico, Mexico, is a federal entity localized in a semiurban area and was designed for the care of low incidence and high diagnostic–therapeutic complexity pathologies in low-income populations. Therefore, efficient and effective care was provided to all patients that attended due to COVID-19 problems. Because this was an emerging pathology, different therapeutic alternatives were sought, such as aggressive oxygen therapy, which effectively reduced the risk of mortality. Unlike most countries, those considered developing countries (including Mexico) presented a higher prevalence of mortality in the socioeconomically active population (40–60 years) due to the inherent shortages of the region.

On the other hand, during the pandemic, each population had access to a different health system as they have different intrinsic characteristics (race, diet, and especially the comorbidities present in Mexico), and populations were affected in different ways. When comparing the morbidity and mortality of the Mexican population (Mexico: 89% diabetes, 24% HBP, and 20% chronic obstructive pulmonary disease) with that reported in other countries by the WHO (for example, Belgium—34% solid tumors; India—30% heart injuries; Iran—diabetes with no specific percentage; Italy—68% HBP and 31% type II diabetes; The Netherlands—25% cancer; Scotland—78.4% diabetes; and Spain—43% cardiovascular diseases), significant differences were observed. Interestingly, in this study, and contrary to the data available in world reports, the comorbidities presented in this unit with greater frequency and related to morbidity and mortality were HBP, drug addiction, and alcoholism, as they showed a statistically significant difference.

In the present study, aggressive oxygen therapy was found to be more effective when increasing blood perfusion and decreasing severe hypoxia compared to reports from other studies. However, it was also found that conventional oxygen therapy was not as effective as thought before the pandemic. [61,62].

Furthermore, it was observed that starting with symptomatic treatment for COVID-19 may have beneficial effects, as mentioned in clinical practice guidelines. However, the use of excessive polypharmacy, as well as certain drugs (e.g., metamizole), can increase mortality.

### Strengths and Limitations

The added value of this study is that the HRAEI is one of the few hospitals in Mexico that manages the electronic clinical record, which made it possible to keep an exhaustive and orderly database of the different clinical and socioeconomic characteristics of each patient. In addition, this tool gave greater internal validity compared to the few similar studies published within the country, reducing biases in the data analysis and guaranteeing that these can be used in systematic reviews and meta-analyses. Evidencing common risk factors and differences in the clinical behavior of the disease will allow the development of a worldwide risk calculator.

Although there were different groups with different work activities, the majority of the population attended belonged to a lower economic stratum, which allowed for a more equitable comparative analysis of prehospital medications.

By carrying out the study prior to vaccination and the generation of other COVID-19 variants in Mexico (circulating variants α, β, and γ), a homogeneous clinical assessment could be performed, allowing the appropriate statistical analysis for mortality and survival with these clinical characteristics. During the first two waves, patients presented similar symptoms (headache, cough, fever, myalgias, arthralgias and asthenia–adynamia, ageusia, anosmia, and dyspnea), differing from the later variants (δ and ο). In the latter variants, in addition to the aforementioned clinical features, rhinorrhea and sneeze were added, and a longer evolution time with lower intensity was observed.

As for the work occupation, the HRAEI, being a hospital providing care to the low-income population, provided services to people in a large number of laborious activities. The existence of different groups meant that the population studied in each of them was reduced.

On the other hand, there were inconveniences in carrying out the interrogation on the prehospital treatment for the care of COVID-19 and its comorbidities. In the case of patients without a nuclear family or primary caregiver, the accompanying persons who admitted the patient did not have the necessary information.

Likewise, during the pandemic, hospitals were overcrowded, resulting in a lack of space in the emergency and intensive care areas, as well as an absence of supplies and equipment for oxygen therapy in an adequate manner for each patient. These complications prevented the correct management of inhalation therapy from admission, directly impacting mortality and survival.

## 5. Conclusions

The main risk factors for COVID-19 mortality were age (group of 40 to 60 years), the male sex, patients with HBP, drug addiction, and alcoholism, as well as occupations such as construction workers, manufacturing workers, carriers, and patients using prehospital medications such as metamizole sodium. The survival factors for this disease were mainly related to healthcare personnel and the use of prehospital medications such as paracetamol, dexamethasone, and azithromycin. It should be noted that aggressive oxygen therapy during hospital admission decreases mortality risk.

## Figures and Tables

**Figure 1 healthcare-11-00932-f001:**
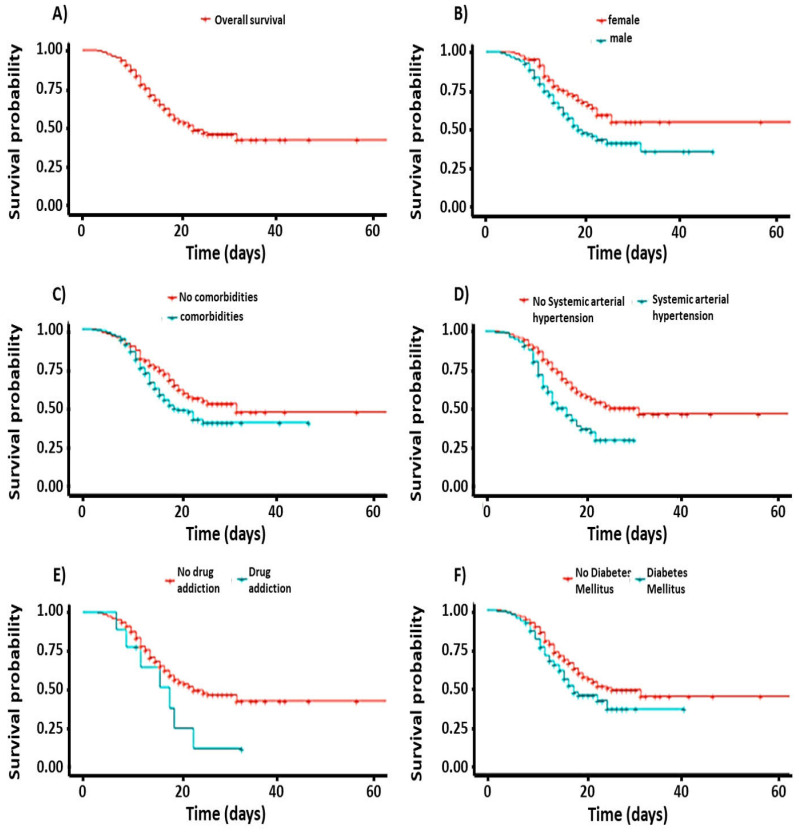
The overall survival curves and medians of subjects with moderate and severe pneumonia due to COVID-19 are presented in this image (**A**). Also shown are the survival curves associated with sex (**B**), the presence of comorbidities (**C**) and HBP (**D**), drug addiction (**E**), diabetes mellitus (DM) (**F**), cough (**G**), hyporexia (**H**), dyspnea (**I**), asthenia and adynamia (**J**), metamizole sodium use (**K**), and steroid use (**L**). Data were analyzed with a Kaplan–Meier curve, and statistically significant associations are assumed with a *p* < 0.05; in addition, 95% confidence intervals are presented for the median of each group (95% CI).

**Table 1 healthcare-11-00932-t001:** Association between sex, occupation, comorbidities, clinical picture, drugs, and in-hospital oxygen therapy according to mortality/survival due to COVID-19.

Variable	Alive	Dead	OR	95% CI	*p*-Value
Sex					
Female	96	43	0.47	0.30–0.72	*p* < 0.05 *
Male	142	135			
Occupation					
Home	47	24	0.51	0.30–0.82	*p* < 0.05 *
Construction worker	3	8	5.22	1.37–2.55	*p* < 0.05 *
Unemployed	4	6	2.93	0.76–1.24	*p* = 0.06
Carrier	12	14	2.28	0.91–5.79	*p* = 0.07
Healthcare personnel	15	2	0.26	0.03–1.02	*p* = 0.09
Comorbidities					
present	93	92	1.66	1.12–2.47	*p* < 0.05 *
absent	145	86			
High blood pressure					
present	42	57	2.19	1.39–3.49	*p* < 0.05 *
absent	196	121			
Alcoholism					
present	49	49	1.58	1.00–2.51	*p* < 0.05 *
absent	192	192			
Drug addiction					
present	2	7	4.83	1.15–32.68	*p* = 0.05
absent	236	172			
Diabetes mellitus					
present	50	49	1.42	0.90–2.24	*p* = 0.12
absent	188	129			
Headache					
present	143	108	0.9	0.83–0.96	*p* < 0.05 *
absent	95	70			
Arthralgias					
present	168	115	0.91	0.85–0.97	*p* < 0.05 *
absent	70	63			
Myalgias					
present	172	118	0.92	0.86–0.98	*p* < 0.05 *
absent	66	60			
Ageusia					
present	25	10	0.87	0.81–0.93	*p* < 0.05 *
absent	213	168			
Dyspnea					
present	229	175	0.93	0.87–1.00	*p* = 0.06
absent	9	3			
Paracetamol					
present	100	57	0.65	0.43–0.97	*p* < 0.05 *
absent	138	121			
Sodium metamizole					
present	18	31	2.57	1.40–4.86	*p* < 0.05 *
absent	220	147			
Azithromycin					
present	190	146	0.47	0.30–0.74	*p* < 0.05 *
absent	48	32			
Steroids					
present	67	32	0.55	0.34–0.89	*p* < 0.05 *
absent	171	146			
Ceftriaxone	65	41	0.63	0.42–0.92	*p* < 0.05 *
Dexamethasone	57	28	0.49	0.30–0.76	*p* < 0.05 *
Enoxaparin	17	8	0.47	0.19–1.05	*p* = 0.07
In-hospital oxygen therapy					
Nasal tips	58	9	0.15	0.07–0.29	*p* < 0.05 *
Reservoir mask	161	165	6.6	3.32–1.46	*p* < 0.05 *

Data were analyzed with a xi^2^. Statistically significant associations are assumed with a *p* < 0.05 *; in addition, 95% confidence intervals are presented for the mean of each group (95% CI) and the odds ratio (OR) for each variable.

**Table 2 healthcare-11-00932-t002:** Associations between mortality/survival due to COVID-19 and age, total hospital stays, stays per service, days of progression, the duration of symptomatology, prehospital treatment dose and length, and the time of intrahospital oxygen therapy.

Risk Factor	Mean		OR	95% CI	*p*-Value
	Alive	Deceased			
Age	45.4	49.1	1.04	1.02–1.07	*p* < 0.05 *
Total stay (days)	9.56	4.6	0.75	0.69–0.79	*p* < 0.05 *
Urgent care stay (days)	0.7	0.8	2.32	1.27–4.43	*p* < 0.05 *
In-hospital stay 1 (days)	8.6	3.6	0.71	0.66–0.77	*p* < 0.05 *
Days of evolution	10.5	8.5	0.89	0.84–0.94	*p* < 0.05 *
Duration of symptoms					
Headache	9.1	7.5	0.9	0.83–0.96	*p* < 0.05 *
Arthralgias	9.5	8.1	0.91	0.85–0.97	*p* < 0.05 *
Myalgias	9.4	8.2	0.92	0.86–0.98	*p* < 0.05 *
Dynamia and/or asthenia	9.4	7.6	0.87	0.81–0.93	*p* < 0.05 *
Dyspnea	3.5	2.9	0.93	0.87–1.00	*p* = 0.06
Oseltamivir dosage	129	220	1.02	1.01–1.06	*p* < 0.05 *
Length of use					
Paracetamol	6	4.8	0.85	0.72–0.99	*p* = 0.05 *
Azithromycin	2.9	4	0.74	0.55–0.96	*p* < 0.05 *
In-hospital oxygen therapy duration					
Device 1	4.9	2.6	0.72	0.65–0.79	*p* < 0.05 *
Device 2	3.42	2.17	0.91	0.84–0.98	*p* < 0.05 *

The days of progression were counted from the beginning of the clinical onset to the hospital admission, the days of total stay were estimated from the admission to the hospital discharge, the urgent care stay was considered from the hospital admission date to the internal medicine (IM) or intensive care (ICU) admission date, hospital stay 1 was considered from the urgent care transfer to the ICU or IM for each individual or until hospital discharge. Data were analyzed with multivariate logistic regression and Student’s *t* test, assuming statistically significant associations with a *p* < 0.05 *; in addition, 95% confidence intervals are presented for the mean of each group (95% CI) and the Odds ratio (OR) for each variable.

## Data Availability

Not applicable.

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
