# Peer review of "Mortality and Survival Factors in Patients with Moderate and Severe Pneumonia Due to COVID-19"

_healthcare, 2023, doi:10.3390/healthcare11070932_

Round 1
Reviewer 1 Report
Great work! The only thing I can suggest is to add a brief description of your method to the abstract.
Author Response
We are thankful for the time and effort you have invested in the revision of our manuscript. All your suggestions have enriched our work. In the manuscript, the additions are highlighted in yellow. Please find our answers to your valuable recommendations; we hope that we have addressed all your concerns.

Reviewer 2 Report
the article is focused on identify the mortality and survival factors in adults with moderate and severe pneumonia due to COVID-19 during the first and second waves of the pandemic, through the analysis of sociodemographic characteristics, symptoms, body mass index, comorbidities, prehospital treatment, and biochemical determinations upon hospital admission. The topic is consistent with the purpose of the magazine.
I suggest specifying the purpose of the research also in the abstract where it is not clear. As far as the introduction is concerned, it appears sufficient, but can also be integrated with other sources (doi: 10.3390/healthcare9020119).
In the material and method section authors can better the describe the hospital (type of hospital, n. of access, n. beds and so on) in order to better identity the scenario of the study.
The discussion is well written I think Authors can compare different parameters and international point of view (I suggest to read and include doi:
10.3390/ijerph18168858Author Response
We appreciate the time and effort you have invested in the revision of our manuscript. Indeed, all your suggestions have improved the quality of our manuscript. In the main text, the additions are highlighted in yellow. We hope that we have correctly addressed all your concerns.

Reviewer 3 Report
Dear Authors
The study is methodologically well done and presents valid and important conclusions. Some minor aspects should be corrected.
The authors should include the OR values in the abstract for the missing variables: The risk factors found were age (95% CI = 47.8-50.3) - (Included the risk factors).
The material and methods must include the city and country: High Specialty Regional Hospital of Ixtapaluca (HRAEI).
The results must include the minimum and maximum age of the subjects. Only the average (47 years) is presented.
In the results appear the occupations as follows: Likewise, the forms of work or occupations were identified as merchants at 22.8%, housewives at 17%, police and general employees at 15.9% each, carriers at 6.3%, and healthcare personnel at 4%.
Subsequently, the occupations referred to as "As for the occupation, ordered by greatest risk, the results are as follows: construction workers with an OR= 5.22 (95% CI [1.37-2.55]". The classification of occupation should be mentioned in the material and methods because it is confusing how the classification was done.
In Table 1, the variables should not be written in capital letters., please, replace it.
Author Response
We appreciate the time and effort you have invested in the revision of our manuscript. Indeed, all your suggestions have improved the quality of our manuscript. In the main text, the additions are highlighted in yellow. We hope that we have correctly addressed all your concerns.

Reviewer 4 Report
Dear authors, this is a well-written study. My main concern is the lack of novelty (there are many published papers describing the mortality in COVID-19). In my opinion, a national journal can be a proper showcase for such a paper, that is why I recommend rejection
Author Response
We appreciate the time and effort you have invested in the revision of our manuscript. We hope to have improved the manuscript by attending to the comments made by the academic editor and the other reviewers. In the main text, the additions are highlighted in yellow.

Round 2
Reviewer 4 Report
dear authors,
I believe this study still has the same weaknesses I mentioned in the first round of reviews.
Age, sex, and comorbidities are well-known risk factors for severe COVID-19 and mortality.
You also tested the influence of work on mortality, but the sample is very little (11 construction workers, 10 unemployed, and 17 healthcare personnel), so your results have poor reliability.
I still think that there are many studies concerning the same topic with larger samples and higher overall quality.
Author Response
4.1 Strengths and limitations
The HRAEI has an electronic health record, which made it possible to keep an exhaustive and orderly data base of the different clinical and socioeconomic character-istics of each patient.
Although there were different groups with different work activities, the majority of the population attended belonged to a lower economic stratum, which allowed a more equitable comparative analysis of prehospital medications.
By carrying out the study prior to vaccination and the generation of other Covid-19 variants in Mexico (circulating variants α, β and γ), a homogeneous clinical assessment could be performed, allowing the appropriate statistical analysis for mor-tality and survival with these clinical characteristics. During the first two waves, pa-tients presented similar symptoms (headache, cough, fever, myalgias, arthralgias and asthenia-adynamia, ageusia, anosmia and dyspnea), differing from the later variants (δ and ο). In the latter variants, in addition to the aforementioned clinical features, rhi-norrhea and sneeze were added, as well as longer evolution time with lower intensity was observed.
As for the work occupation, the HRAEI, being a hospital providing care to the low-income population, provided services to people in a large number of laboral activi-ties. The existence of different groups meant that the population studied in each of them was reduced.
On the other hand, there were inconveniences in carrying out the interrogation on the prehospital treatment for the care of Covid-19 and its comorbidities. In the case of patients without a nuclear family or primary caregiver, the accompanying persons who admitted the patient did not have the necessary information.
Likewise, during the pandemic, hospitals were overcrowded, resulting in a lack of space in the emergency and intensive care areas, as well as an absence of supplies and equipment for oxygen therapy in an adequate manner for each patient. These complica-tions prevented the correct management of inhalation therapy from admission, directly impacting mortality and survival.